

# Experiments of the efficacy of tree ring blue intensity as a climate proxy in central and western China

Yonghong Zheng[1,2], Huanfeng Shen[1], Rory Abernethy[2], Rob Wilson[2]

[1]School of Resource and Environmental Sciences, Wuhan University, Wuhan 430079, China

[2]School of Earth and Environmental Sciences, University of St. Andrews, St. Andrews, KY16 9AL, UK

*Correspondence to*: Rob Wilson (rjsw@st-andrews.ac.uk)



**Abstract**

To investigate the potential value of blue intensity as a robust climate proxy in central and western China, 4 species from 5 sites were assessed. As well as latewood inverted BI (LWB$_{inv}$), we also examined earlywood BI (EWB). To explore the sensitivity of using different extraction parameter settings using CooRecorder, seven percentile variant settings for EWB and LWB$_{inv}$ were used; F50:50 to F95:05. The RW, EWB, and LWB$_{inv}$ were detrended using an age-dependent spline. Correlation

analysis was applied between the tree ring parameter chronologies and monthly/seasonal variables of mean temperature, precipitation, and scPDSI. Linear regression was also used to further highlight the potential of developing climate reconstructions using these species. Only subtle differences were found between the different percentile extraction variants. However, the analysis suggested that F80:20 or F85:15 variants marginally provided better performance. As has been shown for many other northern

hemisphere studies, inverse latewood intensity expresses a strong positive relationship with growing season temperatures (the two southern sites explaining almost 60% of the temperature variance when combined). However, the low latitude of these sites shows exciting potential for regions south of 30ºN that are traditionally not targeted for temperature reconstructions. EWB also shows good potential to reconstruct hydroclimate parameters in some humid areas.




## 1 Introduction

Tree-ring blue intensity (BI), also sometimes called blue reflectance, was initially explored as a
substitute for maximum latewood density (MXD) and has been shown to express similar
dendroclimatic potential as density parameters and is relatively inexpensive and easy to produce

(Yanosky and Robinove, 1986; Björklund et al., 2015; Björklund et al., 2019; Reid and Wilson, 2020;
Kaczka et al., 2018; Wilson et al., 2014).  Sheppard et al. (1996) first confirmed that reflected-light
image analysis could provide a substitute for X-ray densitometry for dendroclimatology, and derived
the first reflected light based temperature reconstruction. These earlier studies (Sheppard et al., 1996;
Yanosky and Robinove, 1986) used video-camera-based systems for image capture. McCarroll et al.

(2002) later showed that a scanner-based system could be used to capture suitable digital images and
assessed the suitability of mean, maximum, and minimum reflectance values for red, green, blue visible
light, as well as ultraviolet bands by correlating the reflectance data with maximum density, which
showed that minimum blue reflectance was the most robust proxy measure of latewood density.
McCarroll et al. (2002) proposed that the minimum blue light reflectance measured the amount of light

absorbed by lignin in the latewood cell walls. Campbell et al. (2007, 2011) advanced the scanner-based
system method (Mccarroll et al., 2002) by avoiding reliance on specialist image analysis software and
utilized the commercial and widely used software WinDENDROTM to confirm that minimum blue
intensity measurements from resin extracted Scots pine laths provided a robust and reliable surrogate
for maximum density and summer temperatures. Compared to WinDENDRO[TM], a lower cost

alternative for measuring BI has been incorporated into the CooRecorder/CDendro software package,
by which several experiments (Rydval et al., 2014; Wilson et al., 2012; Wilson et al., 2014) have been
conducted with this approach now becoming more and more popular in the tree ring community
(Kaczka and Wilson, 2021).

BI-based tree ring research, focusing on both climate and ecological based studies, have been widely

carried out in Europe (Helama et al., 2013; Babst et al., 2009; Mccarroll et al., 2002; Campbell et al.,
2007; Rydval et al., 2014; Dolgova, 2016; Fuentes et al., 2018) and North America (Wilson et al., 2014;
Wiles et al., 2019; Harley et al., 2021; Heeter et al., 2021; Wilson et al., 2019; Wang et al., 2020).
Recently, some attempts have been made to explore the utility of BI for dendroclimatology in Australia
(Wilson et al., 2021; Brookhouse and Graham, 2016; Blake et al., 2020; O'connor et al., 2022) and Asia



(Buckley et al., 2018; Cao et al., 2022; Davi et al., 2021). As the biggest territory in Asia, China has several types of climates due to different geographical zones which provides a golden opportunity to conduct BI based dendroclimatic experimental research. To date, tree ring metrics, such as tree ring width (RW), stable isotopes and density have been used in are very unbalanced way in China. A recent review (He et al., 2019) on advances in dendroclimatology in China, showed that tree-ring width, stable

oxygen, stable carbon, and density account for 73%, 13%, 7%, and 7% of all reviewed chronologies from China respectively, with BI not being mentioned at all. In fact, BI based dendroclimate research is extremely rare (Cao et al., 2022; Cao et al., 2020) in China to date. It is obvious that there are significant gaps and opportunities for BI based dendroclimate research in China.

Building on Rydval et al. (2014), which provided a methodological guide for the generation of BI data

using CooRecorder, we present here extended experiments exploring the sensitivity of using a range of percentile extraction parameterizations for both dark (latewood) and light (earlywood) pixels for BI data generation. Our study utilizes samples from 4 conifer species from western and central China (Fig.1) and assesses the potential of these species for BI based dendroclimate research.

**Figure 1**

**2 Materials and methods**

**2.1 Study location and sample information**

For this study, increment cores were taken between 2013 and 2021 for 4 coniferous tree species from 5 sites across China (Table 1). *Picea crassifolia* from Wulan county (TL) and Xiariha (XRH) in Dulan county of Qinghai province, *Abies fargesii* from Jinhouling (JHL) of Shennongjia Mountain in Hubei

province, *Picea likiangensis* and *Abies fargesii var.faxoniana* from Yulong snow Mountain (YL) and Laojunshan Mountain (LJS) in Yunnan province.

**Table 1**

The climatological context of the sampled sites is very diverse. Using the CRU TS4.05 (Harris et al., 2020) climate data grid (1991-2020), annual mean temperatures for TL, XRH, JHL, YL, and LJS are -

3.43 °C, 2.34°C, 15.40°C, 6.15°C, 7.28°C, while total annual precipitation is 203.78 mm, 265.05 mm, 1041.24 mm, 870.14 mm, and 935.00 mm respectively (Fig.2). The sites therefore represent a range



from high elevation cold and dry sites (e.g. TL and XRH, are located in a high elevation arid plateau climatic region) to lower elevation warm and humid locations (JHL).

**Figure 2**

**2.2 Tree ring data**

Our samples, including spruce and fir, do not express a visible heartwood-sapwood color change and so no pre-treatment (i.e. resin extraction) was performed (Dolgova, 2016; Wilson et al., 2019). The mounted cores were sanded from 240 to 800 grit grade before being scanned with a flatbed Epson V850 Pro scanner. The scanner was calibrated using the SilverFast scanner software to the IT8 color card Target (IT8.7/2) printed on Kodak Professional Endura paper. This calibration step is important to

ensure consistency between labs as well addressing the potential temporal instability in the power or intensity of the light because bulbs tend to fade over time, leading to a potential drift in blue intensity values (Campbell et al., 2011).

All tree ring samples were scanned at 3200 DPI with the scanner covered by a box (with matt black

side walls) to minimize bias from external ambient light and internal box reflections of light. The scanned digital image of each sample was then imported into CooRecorder and the ring-boundaries marked by both manual and automatic placement (Maxwell and Larsson, 2021). COFECHA (Grissino-Mayer, 2001) was utilised to validate the reliability of the tree ring dating. Inverted latewood blue intensity ($LWB_{inv}$) (Rydval et al., 2014) data were generated using frame specification parameters

controlling the "window" from which reflectance measurement were derived (width-offset-limiting-depth-margin, 300-3-5-500-0.5). Earlywood blue intensity (EWB) data were generated using frame specifications (200-3-0-500-0.5). A range of percentile values for EWB and $LWB_{inv}$ were used to extract different light and dark wood reflection intensity information, including 50-50, 60-40, 70-30, 80-20, 85-15, 90-10, 95-5. This novel approach was explored to test whether there is a methodological

influence upon the relationship between the variable BI parameters and climate variables for varying percentile extraction options for these parameters. To develop the chronologies, both the ring-width and BI data-sets were detrended using an age dependent spline (Melvin et al., 2007) with the programme ARSTAN, with an initial starting 50-year spline, which more naturally tracks the juvenile and long-term trajectory of radial growth than more rigidly defined approaches such a negative

exponential functions.



**2.3 Climate data**

Considering most meteorological stations were not founded before the 1950s in study areas, monthly climate data for the period 1951-2012, including mean temperature (TMP), precipitation (PRE), and self-calibrating palmer drought severity index (scPDSI) were extracted from the CRU TS4.05 climate

data grid (http://climexp.knmi.nl/) with a resolution of 0.5∘ × 0.5∘. We used the mean value of the four closest gridded points to each sampling site.

**2.4 Data analysis**

To assess the different statistical qualities between the tree-ring variable chronology variants, the coefficient of variation (CV), first order autocorrelation (AC1), and mean inter-series correlation (Rbar)

were evaluated. CV, which is the ratio of the standard deviation to the mean, quantifies the relative variance of the chronologies. It is useful to compare variance between data sets with different units (i.e. ring-width vs. BI) or with widely different means. The higher the CV, the greater the relative dispersion around the mean. AC1 measures the persistence structure in time-series (i.e. the year-to-year correlation of a time-series with itself at lag 1). The higher the AC1, the stronger the relationship

between consecutive years of data. Rbar is the mean inter-series correlation of all possible detrended bivariate pairs of tree ring series in a chronology. The higher the Rbar, the stronger the common signal in the data that makes up the chronology. To further explore the potential of these tree species and variables for dendroclimatic research, correlation analysis was carried out between tree ring chronologies and monthly/seasonal variables of each climate variable using the common time interval

1951-2012 (Table 1). Finally, multiple linear regression was performed for the strongest TR parameter vs climate relationships that are biological most meaningful to highlight the potential for dendroclimatic reconstruction for these tree species.

**3 Results and discussion**

**3.1 Chronology Statistical Properties**

CV is much higher for RW than EWB and $LWB_{inv}$ (Fig.3a), an observation detailed for other studies comparing RW with BI parameters (Wilson et al., 2021; Wilson et al., 2014). The CV values for $LWB_{inv}$ are similar but have a wider spread than EWB. These low relative variance values are one



reason why the signal strength statistics are often weaker for BI parameters than other parameters such as RW and MXD as any non-climatic signal (e.g. wood discoloration) will have a large impact on the

Rbar values (see below).

RW and EWB express similar median AC1 values (Fig.3b) although the YL site expresses substantially lower values resulting in a much wider range for EWB. Overall, LWB$_{inv}$ AC1 values are generally lower, again agreeing with other studies (Reid and Wilson, 2020; Kaczka et al., 2018) assessing both LWB$_{inv}$ and MXD which generally express low 1$^{st}$ autocorrelation for conifers from temperature

sensitive sites. This is a desirable property as LWB$_{inv}$ often correlates strongly with summer temperatures and which also expresses low AC1.

The range in Rbar values between all three parameters is very large (Fig.3c). RW expresses highest overall Rbar values – with the TL RW data showing a very strong common signal, and LJS weakest. EWB and LWB$_{inv}$ express much weaker signal strength, although median values are similar. LWB$_{inv}$

expresses a much greater range than EWB with TL expressing a strong common signal where only about 11 trees are needed to attain an EPS of 0.85. LJS on the other hand shows a very weak common signal where theoretically more than 50 trees are needed to attain an EPS of 0.85 (Wilson and Elling, 2004). The weaker common signal of the BI parameters has been noted in several studies (Wilson et al., 2021; Wiles et al., 2019; Blake et al., 2020; Harley et al., 2021), with both EWB and LWB requiring

greater sample replication than RW to reach widely accepted thresholds of chronology reliability (Blake et al., 2020; Harley et al., 2021). However, as has been shown in several previous studies, the weaker common signal in BI chronologies does not necessarily mean that the climate signal is weaker than RW (Wilson et al., 2019; Rydval et al., 2014).

**Figure 3**

The differences in CV, AC1 and Rbar values are subtle between the different percentile extraction chronology versions (Fig.3 and Supplementary Table 1). There appears to be little consistency as to which of the percentile extraction methods leads to consistent high or low values of CV, AC1 and Rbar. For EWB, highest CV values are noted for the F50:50 variants (F60:40 shows the same value for YL) except for JHL where F70:30 expresses the highest CV value. For LWB, again F50:50 variants express

higher CV values (F60:40 shows the same value for TL and JHL) with LJS deviating away from this with F85:15 to F95:5 showing highest values. For AC1, there appears to be no consistent pattern of




high and low values between each percentile variant for both BI parameters. However, either F50:50 or F95:05, or together with the adjacent percentile variants, express the highest value except for EWB for JHL and LJS. For Rbar, two of the EWB percentile variants express the strongest signal strength for

both F50:50 and F95:5 while for $LWB_{inv}$, the results are equally variable. Overall, the chronology characteristics based on different extraction percentiles vary minimally, suggesting that the percentile extraction settings are not a significant factor for either EWB or $LWB_{inv}$ data generation.

**3.2 Climate response of the chronology variants**

The strength of correlations between the RW chronologies and monthly TMP, PRE, scPDSI vary

substantially across the different sites (Fig.S1). Over the period 1951-2012, TL RW expresses significant positive correlations with scPDSI for January through August (Fig.4a, Fig.S1), which may result from the relative dry conditions indicated by the negative scPDSI values for this location (Fig.2). TL RW vs Jan-Jul scPDSI explains 36.9% of the drought index variance. Except for Jun TMP at LJS, the correlations between RW and climate for XRH, JHL, YL, and LJS, are not significant as the

climatic influence on RW is mixed and hence no reconstruction of past climate, using this parameter, is possible in these regions (Rydval et al., 2016).

**Figure 4**

EWB measures the max intensity values of the light pixels, reflecting the lumen size of the earlywood - i.e. large vacuole and thin cell walls - and so reflects tree ring minimum density. EWB shows varying

response to TMP, PRE, and scPDSI - weakest for TMP and the strongest for scPDSI. For TMP, significant positive correlations are only noted for JHL (April-May) (Fig.S2), which may result from a higher spring TMP promoting tree growth (Zheng et al., 2016). For PRE, late spring or early summer shows significant positive influence at TL, XRH, and YL (Fig.S3). These results are encouraging and fits with recent research in Sweden where it was found that BI based precipitation calibrations can

explain 20% more hydroclimatic variance compared to ring width (Seftigen et al., 2020). scPDSI expresses universal positive influence on EWB at all sites except JHL (Fig.S4). A positive relationship with PRE (Fig.S3) and a negative response to TMP (Fig.S2), indicates that drought conditions are the main limiting factor for the variability of cell wall size (Begović et al., 2020).

For $LWB_{inv}$, although the sample sites are not located near upper tree line, a significant TMP response

is noted for all the sampling sites (Fig.S2), which suggests the possibility to enhance the climate




response of BI chronologies via sampling closer to upper tree line (Heeter et al., 2021). Especially significant is the relationship between $LWB_{inv}$ and August TMP ($r = 0.593$ for F80:20) at JHL (Fig.S2). To eliminate the potential inflation of correlation values due to coherent low frequency trends between time-series, we also calculated the correlation after first differencing both $LWB_{inv}$ and August TMP.

The first differenced correlations are even stronger at $> 0.7$ suggesting that there is some degree of dissimilarity at decadal and longer timescales between BI and the climate data (Wilson et al., 2019; Blake et al., 2020). The positive relation between $LWB_{inv}$ and TMP is analogous to the positive relation between MXD and growing-season temperature (Wilson et al., 2012). The strongest inverse influence shown by PRE on $LWB_{inv}$ are identified at comparatively humid sites JHL and YL, which fits in with

the positive temperature response of $LWB_{inv}$ with TMP and the inverse correlation between PRE and TMP. Though the correlations between $LWB_{inv}$ and scPDSI are relatively weak, significant positive correlations with scPDSI between January and April are noted for TL with inverse correlations for the autumn with YL.

We utilized the single month correlation function analysis (Fig.S2-S4) and systematic correlation

analysis results (Fig.4) to identify the optimal, and biologically most relevant, single month or seasonal window to maximize the TR parameter and climate relationships. We then use this single month or season to test how the correlation value between these optimized relationships changed for the different percentile variants. Overall, there is no one single percentile combination for EWB and $LWB_{inv}$ that stands out for those monthly and seasonal relationships that express the strongest correlations (Fig.5).

Although the differences are subtle, in most cases except for LJS $LWB_{inv}$ vs May-Oct TMP (Fig.5e), using 50:50, 60:40, and 70:30 do not return optimal results (Fig.5). However, 80:20 and 85:15 generally provide the strongest results, a similar result to earlier experiments from Scotland (Rydval et al., 2014). However, as more higher resolution methods are employed for image capture, we urge the community to continue experimenting with varying percentile extraction settings to help provide a

theoretical basis for optimal settings.

**Figure 5**

$LWB_{inv}$, which has proven to be a robust proxy for summer temperature at high northern latitudes reconstruction TMP (Harley et al., 2021), also express very strong TMP signals for the mid-to-low latitude (Heeter et al., 2021). However, most BI studies to date are still primarily geographically



restricted to the high latitudes. More studies are needed to evaluate the applicability of BI methods

across different regions, especially at high-elevation, low-latitude locations, where certain tree species

still produce distinct annual growth rings (Heeter et al., 2020). The lower latitude sites, including JHL,

YL, and LJS in central and southwest China, with a collective strong TMP signals response in $LWB_{inv}$

(Fig.4 and Fig.S2), show great potential to reconstruct past temperatures for these relatively lower

latitudes. The cool and rather humid climate regime of YL, a site type which traditionally has been

overlooked in tree ring studies for hydroclimate, shows great potential when using EWB due to the

strong implicit hydroclimatic signal expressed in scPDSI (Fig.4 and Fig.S4). This observation, along

with similar results for southern Sweden confirm the importance of both EWB and $LWB_{inv}$ for

understanding of hydroclimate variability in regions with a humid climate (Seftigen et al., 2020).

Finally, we use stepwise multiple regression of multi-site-parameter BI data to highlight the

improvement, by using data from multiple sites, over these single parameter results (Fig.6). Focusing

on regionally grouped data-sets, EWB data from XRH and Tl, and LJS and YL explain 27% and 30%,

respectively, of the June-July scPDSI variance. Although these results are modest, we hypothesize that

expanding the number of sites and including RW data would result in substantially improved PDSI

reconstructions for this region. Further, by also using the $LWB_{inv}$ data from LJS and YL, the multiple

regression combination of these data results in extremely strong calibration with June-October

temperatures ($R^2$ adj = 0.59, Fig.6), with the reconstruction representing a large region of low latitude

China. These results demonstrate the considerable potential of using BI to enhance current RW-based

climate reconstructions in China.

**Figure 6**

**4 Conclusions**

In this study, we measured RW, EWB, and $LWB_{inv}$ for 5 sites in western and central China to

investigate the potential value for BI variables to enhance dendroclimate research. We have focused on

species (*Picea and Abies*) that express no visible color change from the heartwood to sapwood so

minimizing trend biases in the BI data. We further explored how sensitive the results are to different

percentile extraction parameter settings for attaining blue intensity data using CooRecorder.



The results presented herein, strongly indicate that BI parameters will enable a significant improvement upon RW based dendroclimatic reconstructions in China. Perhaps the most compelling factor of the BI method is that tests can be easily and quickly made on multiple samples, sites, and species from

varying locations, so a broader picture of the potential of measuring multiple tree-ring parameters from many species can be easily tested. Our results indicate and agree with most other northern hemisphere (NH) studies exploring conifer response to climate (Rydval et al., 2014; Heeter et al., 2021), that $LWB_{inv}$ expresses a positive relationship with growing season temperatures. Despite data from only two sites, the combined information from sites LJS and YL explain almost 60% of the temperature

variance which is on par with some of the strongest calibrations noted in the NH (Wilson et al., 2016). However, these results are particularly exciting due to the low latitude location of these sites where traditionally, temperature reconstructions are poorly constrained at latitudes south of 40oN (Anchukaitis et al., 2017; Wilson et al., 2016). We hypothesize that these results would improve by sampling more trees and sampling more sites closer to the upper tree-line.

EWB is still a relatively untested parameter in dendroclimatology. Our experiments strongly suggest that this parameter could greatly enhance reconstructions of past hydroclimate, especially PDSI, over those relying solely on RW data.

Although experiments using different percentile extraction parameters for EWB and $LWB_{inv}$ did not identify a clear optimal set of settings for the BI data extraction, F80:20 and F85:15 are recommended

due to their good performance in most cases (Fig.5). However, we encourage the community to continue further experimentation with different data extraction parameterization in CooRecorder, as our current results were produced from scanned images. It is entirely possible that as labs start experimenting with higher resolution image capture methods (Levanič, 2007), different extraction parameters may be needed to improve climate response.

The challenge now is to expand the network of Chinese BI chronologies with more species and locations, but also identify preserved wood sources that will allow a significant extension back in time. Finding older stands of trees is of course a priority, but that is not always possible in regions where humans have lived for a significant length of time. The focus must therefore be on extending the shorter living chronologies using preserved material from historic buildings (Wilson et al., 2004) or

natural environments where wood is preserved such as in anoxic lake sediments.



**Data availability**

All raw data for pictures and tables can be provided by the first author upon request.

**Author contribution**

YZ, HS and RW planned the campaign; YZ and HS sampled the tree-ring samples; YZ and RA
performed the measurements; YZ analyzed the data and wrote the manuscript draft; RW reviewed and
edited the manuscript.

**Competing interests**

The contact author has declared that neither of the authors has any competing interests.

**Disclaimer**

Publisher's note: Copernicus Publications remains neutral with regard to jurisdictional claims in
published maps and institutional affiliations.

**Acknowledgements**

We thank Penghai Wu, Shuqiang Meng, Ziwen Zhao, and Zhengsheng Hu for helping to sample tree-
ring samples.

**Financial support**

This paper was funded by the China Scholarship Council No.202006275018, and the National Natural
Science Foundation of China (NSFC) Project No. 41771227.
RW was further funded on the NSF/NERC project (NE/W007223/1) – Understanding Trans-
Hemispheric Modes of Climate Variability: A Novel Tree-Ring Data Transect spanning the Himalaya
to the Southern Ocean.

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




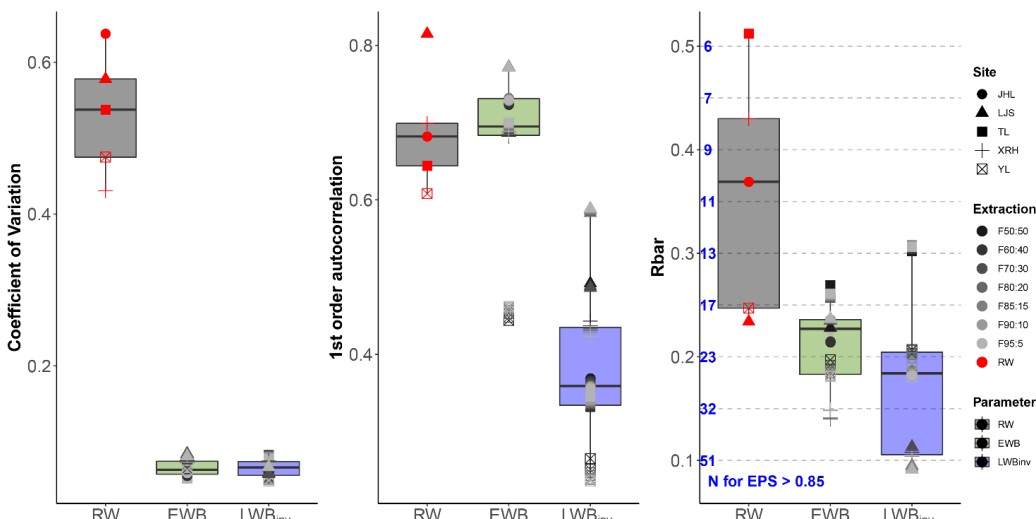

**Fig.3 CV, AC1, and Rbar for each standard chronology – delineated by parameter, site, and BI percentile extraction**

460

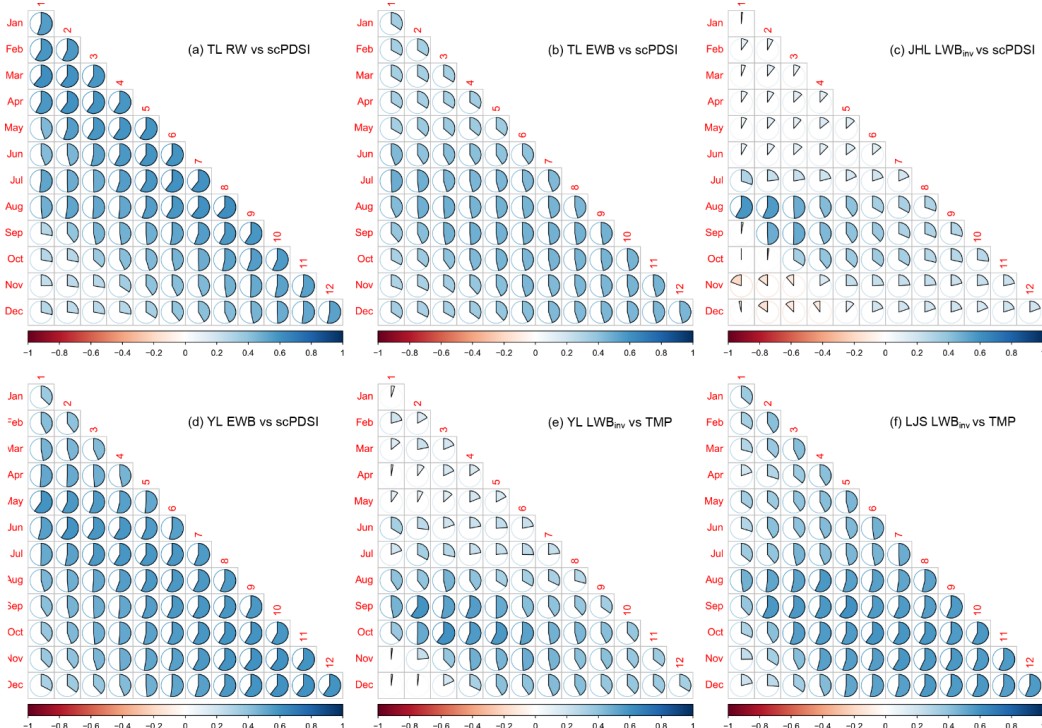

**Fig.4 Correlation analysis of select parameter chronologies against different climate targets for different end months (along the y-axis) and different season lengths (the number along**





**the diagonal line). Both the ratio & color of the shaded portion of the pie denotes the**
465          **correlation coefficient. We show F70:30 variants for these examples**

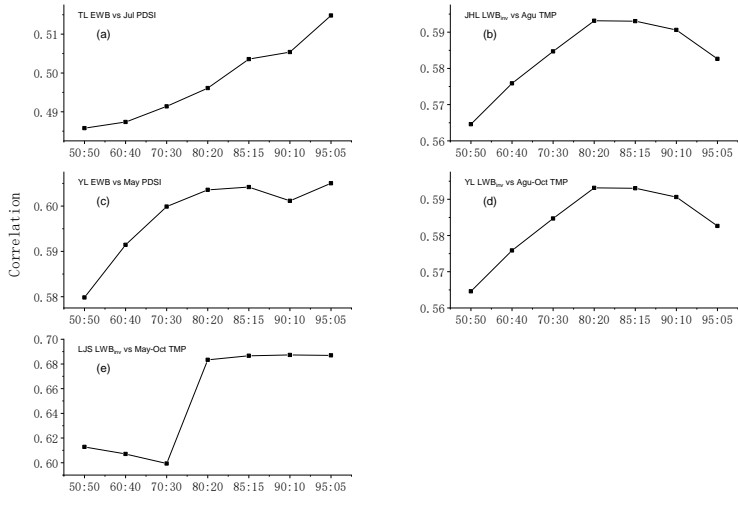

**Fig.5 Correlations for different percentile extraction variants for those parameter**
470          **chronologies and climate variables that express the strongest signal**

.





**Fig.6 Experimental multiple regression calibration (1951-2012) results using multi-site regression models for XRH, Tl, LJS and YL. F85:15 variants used.**



475

**Table 1 Sample information**

| Site code | Species | Climate Zone | Elevation (m) | Hight below tree line (m) | Vertical distribution range (m) | Cores | Full Period |
|---|---|---|---|---|---|---|---|
| TL | *Picea crassifolia* | Plateau climatic region | 3700 | 100 | 2600-3800 | 34 | 1821-2014 |
| XRH | *Picea crassifolia* | Plateau climatic region | 3720 | 80 | 2600-3800 | 44 | 1907-2014 |
| JHL | *Abies fargesii* | North subtropical zone | 2564 | 541 | 2000-3105 | 69 | 1830-2021 |
| YL | *Picea likiangensis* | Mid subtropical zone | 3377 | 823 | 3100-4200 | 35 | 1936-2013 |
| LJS | *Abies fargesii var.faxoniana* | Mid subtropical zone | 3587 | 413 | 3000-4000 | 33 | 1688-2013 |

e: Highest values highlighted using shadow.