# Peer review of "Experiments of the efficacy of tree ring blue intensity as a climate proxy in central and western China"

_EGUsphere, 2023_

## Author Response (AR1)

Dear Editor,

I hope this letter finds you well. We have made significant revisions to the manuscript based on the comments provided by you and the referees. Specifically, we have made the following changes:

- 1. Updated abbreviations: We have revised all the abbreviations "TL" to "WL" throughout the manuscript to ensure consistency.
- Added chronologies to supplement file: We have included the following chronologies in the supplement file: ring width, early wood blue intensity, and latewood inversed blue intensity. Consequently, we have changed the figure number for the supplement file accordingly.
- 3. Revised abstract: The abstract of the manuscript has been revised to reduce the use of abbreviations, making it more accessible to readers.
- 4. Updated tree ring data: We have meticulously checked and updated the original tree ring data to enhance the reliability of the results. As a result, there have been minor changes in some of the findings. All figures and tables, both in the manuscript and supplement, have been updated accordingly.
- 5. References: In order to improve the quality of the manuscript, we have removed references that were deemed unsuitable and added new references that are more relevant to the topic.
- 6. Improved wording: We have carefully reviewed and updated the wording and expressions throughout the manuscript to ensure appropriateness and clarity.

We believe that these revisions have strengthened the manuscript significantly. However, if there are any further modifications that you deem necessary, please do not hesitate to let us know. We appreciate your time and consideration.

Thank you.

Sincerely,

Yonghong

**Comments of Referee 1**

This paper tests the potential of tree ring blue intensity as a climate proxy in China. It showed that tree ring blue intensity expressed a strong relationship with climate, which proved the potential value of tree ring blue intensity in dendroclimatic research. Considering the extensive scope of China and the relatively limited number of tree-ring blue-intensity studies, this research is encouraging and exciting. In particular, it showed great potential to reconstruct past temperatures for relatively lower latitudes. However, there are still some issues with which I am concerned as followed:

Why use inverse latewood blue intensity? If not inversed, what is happen?

It is the same tree species for TL and XRH. Why does the ring width show a different response to climate in Fig.S1?

Why does the weaker common signal of BI parameters compared to tree ring width not limited their utility in keeping climatic signals?

What causes the higher correlation between JHL LWBinv and TMP at a higher frequency than the original series?

In summary, I recommend that the authors address the above concerns. With these concerns addressed, this paper will make a valuable contribution to the tree ring blue intensity research field. Furthermore, it can promote tree ring blue intensity-related studies in Asia and China.

**Reply to Referee 1**

1. Why use inverse latewood blue intensity? If not inversed, what is happen?

This really comes down to detrending and the limitations of using standard methods within Arstan. i.e. negative trends are removed but positive trends are retained etc LWB is also inverted so it "behaves" in a similar way to MXD. In fact, there are some papers that use the original latewood blue intensity data in the early years.

2. It is the same tree species for TL and XRH. Why does the ring width show a different

response to climate in Fig.S1?

The same tree species was used for TL and XRH, but it exhibited different climate responses, likely due to the varying climates observed in TL and XRH as shown in Fig. 2. One noticeable difference is the month with the highest precipitation, which differs between the two sites. Overall, there are differences in the strength of the climate response between the two sites, but there is considerable consistency in the way they respond.

3. Why does the weaker common signal of BI parameters compare to tree ring width not limited their utility in keeping climatic signals?

The weaker common signal and weak EPS does not necessarily mean that the climate signal will be weak – as has been shown for many studies now. The implications are likely that for future studies, we simply need to sample/process more trees which will improve the calibrated signal a little.

4. What causes the higher correlation between JHL LWBinv and TMP at a higher frequency than the original series?

Again – this was an observation noted in earlier BI studies – this again comes down to the common signal and EPS issue – the low frequency is more impacted by discoloration issues, so where the high frequency generally behaves well, it takes more samples to attain a more robust low-frequency signal. The same concept goes for using RCS detrending for RW or MXD etc.

**Comments of Referee 2**

This manuscript investigated the potential value of blue intensity (BI) in central and western China based on latewood and early wood BI 4 species from 5 sites. Different percentile variant settings were used to test the sensitivity to the results. Correlation with climatic factors showed that the inverse latewood intensity from the two southern sites is a good indicator of growing season temperatures but earlywood BI has potential to reconstruct hydroclimate variations in some wet areas. This manuscript is short but intriguing. It is a comprehensive assessment of blue intensity as a climate proxy in China although the number of site and species are small. I have some question before it goes forward.

4 coniferous tree species from 5 sites across China are used in this study. They are Picea crassifolia from Wulan county (TL) and Xiariha (XRH) in Dulan county of Qinghai province, Abies fargesii from Jinhouling (JHL) of Shennongjia Mountain in Hubei province, Picea likiangensis and Abies fargesii var.faxoniana from Yulong snow Mountain (YL) and Laojunshan Mountain (LJS) in Yunnan province. How about the efficacy of the Picea crassifolia BI from Wulan and Xiariha? Since this region is characterized by cold and wet, it is expected that the BI is much more represented for temperature than in southern China. A small question: Wulan county, the abbreviation is TL, why not WL?

Figure 4: Why did not it present the results from TL LWBinv with climate factors?

In general, the northern TP has developed many long chronologies, including ring width and isotopes chronologies which reflect precipitation variations. However, the tree lifespan of the studied species is short in southern China. What new gains could be obtained from the blue intensity compared with traditional ring width parameter? It would much better to show the whole chronologies of different parameters over time.

**Reply to referee 2**

1. This manuscript investigated the potential value of blue intensity (BI) in central and western China based on latewood and early wood BI 4 species from 5 sites. Different percentile variant settings were used to test the sensitivity to the results. Correlation with

climatic factors showed that the inverse latewood intensity from the two southern sites is a good indicator of growing season temperatures but earlywood BI has potential to reconstruct hydroclimate variations in some wet areas. This manuscript is short but intriguing. It is a comprehensive assessment of blue intensity as a climate proxy in China although the number of site and species are small. I have some question before it goes forward.

Reply: We agree that the number of sites is small, but this initial paper was designed as an explorative assessment of Blue Intensity for some commonly used species in China.

2. 4 coniferous tree species from 5 sites across China are used in this study. They are Picea crassifolia from Wulan county (TL) and Xiariha (XRH) in Dulan county of Qinghai province, Abies fargesii from Jinhouling (JHL) of Shennongjia Mountain in Hubei province, Picea likiangensis and Abies fargesii var.faxoniana from Yulong snow Mountain (YL) and Laojunshan Mountain (LJS) in Yunnan province. How about the efficacy of the Picea crassifolia BI from Wulan and Xiariha? Since this region is characterized by cold and wet, it is expected that the BI is much more represented for temperature than in southern China.

Reply: For the main paper figures, we have focused on the strongest relationships between the TR parameters and climate. However, we detail the relationships of the TL and XRH LWB chronologies with temperature in Figure S2. There is some weak response for May (TL) and Aug (XRH) but the correlations are not strong enough to consider any robust climate modeling. It seems surprising that the high-elevation sites of LWB of TL and XRH do not reflect a summer temperature response. However, this lack of temperature response can be attributed to the limited precipitation in these areas, which has a stronger effect on local tree growth. Therefore, the tree-ring parameters from these sites may be more indicative of water availability than temperature. It may be worth considering this factor when interpreting the paleoclimatic signals obtained from these sites.

3. A small question: Wulan County, the abbreviation is TL, why not WL?

Reply: WL may be one of the suitable abbreviations for this site. However, we chose to use TL as an abbreviation for Wulan County because the sampling site is located near a railway (called Tie Lu in Chinese).

4. Figure 4: Why did not it present the results from TL LWBinv with climate factors?

Please see the earlier comment. These results are presented in supplementary figure S2. The relationship is not strong enough to go further than standard response functions. To keep the paper reasonably short, we only focus on the strongest results in the main paper.

5. In general, the northern TP has developed many long chronologies, including ring width and isotopes chronologies which reflect precipitation variations. However, the tree lifespan of the studied species is short in southern China. What new gains could be obtained from the blue intensity compared with traditional ring width parameter? It would much better to show the whole chronologies of different parameters over time.

Reply: The lifespan of the studied tree species is shorter than that of some other trees, such as Juniperus przewalskii Kom. Despite this, exploring the use of tree-ring blue intensity as a climate proxy is valuable, as different tree species may contain different climate signals. In our research area, the EWB of TL and XRH showed a strong climate signal of scPDSI, compared to other tree-ring parameters. In this paper, we did not show the full chronologies for the example sites due to the large number of variants for EWB and LWBinv chronologies, as well as the number of figures in the paper. However, we aim to identify close relationships between blue intensity and climate using common years for chronologies and climate first, before expanding the chronologies for those with strong climate signals in future work. These findings could have significant implications for understanding climate variability and change in the study region.